# Comparison of Remote Sensing Techniques for Geostructural Analysis and Cliff Monitoring in Coastal Areas of High Tourist Attraction: The Case Study of Polignano a Mare (Southern Italy)

**Lidia Loiotine** [1,2,*] **, Gioacchino Francesco Andriani** [2] **, Michel Jaboyedoff** [1] **, Mario Parise** [2] **and Marc-Henri Derron** [1]

1   Institute of Earth Sciences, University of Lausanne, 1015 Lausanne, Switzerland; michel.jaboyedoff@unil.ch (M.J.); marc-henri.derron@unil.ch (M.-H.D.)
2   Department of Earth and Environmental Sciences, University of Bari Aldo Moro, 70125 Bari, Italy; gioacchinofrancesco.andriani@uniba.it (G.F.A.); mario.parise@uniba.it (M.P.)
*   Correspondence: lidia.loiotine@unil.ch

**Abstract:** Rock slope failures in urban areas may represent a serious hazard for human life, as well as private and public property, even on the occasion of sporadic episodes. Prevention and mitigation measures indispensably require a proper rock mass characterization, which is often achieved by means of time-consuming, costly and dangerous field surveys. In the last decades, remote sensing devices such as high-resolution digital cameras, laser scanners and drones have been widely used as supplementary techniques for rock slope analysis and monitoring, especially in poorly accessible areas, or in sites of large extension. Although several methods for rock mass characterization by means of remote sensing techniques have been reported in specific studies, there are very few contributions that focused on comparing the different methods in an attempt to establish their advantages and limitations. With this study, we performed digital photogrammetry, Terrestrial Laser Scanning and Unmanned Aerial Vehicle surveys on a cliff located in a popular tourist attraction site, characterized by complex geological and geomorphological settings, as well as by disturbance elements such as vegetation and human activities. For each point cloud, we applied geostructural analysis by means of semi-automatic methods, and then compared multi-temporal acquisitions for cliff monitoring. By quantitative comparison of the results and validation by means of conventional geostructural field surveys, the pros and cons of each method were outlined in attempt to depict the conditions and goals the different techniques seem to be more suitable for.

**Keywords:** digital photogrammetry; Terrestrial Laser Scanning; Unmanned Aerial Vehicle; coastal area; rock mass characterization; slope monitoring; karst; dynamic disturbance

## 1. Introduction

Local to global failures in urbanized steep rock coasts represent a serious threat to the natural landscape, infrastructure and human activities. Cliff retreat is the cumulative result of several marine and subaerial processes acting at different temporal and spatial scales, which were described and simulated by several authors [1–10] in an attempt to understand and predict the geomorphologic evolution of rock coasts. Given the interaction between geo-environmental processes and material properties, hazard assessment and planning of prevention or mitigation measures in coastal areas require a full understanding of the specific site conditions, with particular emphasis on the mechanical behavior of the rock mass, as well as the failure volumes, time frequency and modes. All of the above need to be carefully evaluated in relationships with the human infrastructures and activities present in the area, as a crucial step in the definition of the related risk.

At a preliminary stage, environmental engineering solutions are planned by means of rock mass classification systems that have been proposed since the 1950s in the international

literature [11–15]. At a more detailed level, geotechnical models should be produced: they are 2D or 3D simplified and schematic representations of the geomorphologic, geostructural and hydrogeological setting of the study site, and of the physical and mechanical properties of the rock materials [16–18]. Particular attention should be given in conceptualizing primary (i.e., bedding planes) or secondary (i.e., joints, faults) discontinuities within the models, since they cause anisotropy of rock masses and influence their mechanical behavior [13,19–22]. Numerical simulations are carried out on the geotechnical models to predict the location and kinematics of potential instabilities by adopting the appropriate technique [23–26]. Based on these considerations, the success in numerical modelling and in susceptibility assessment depends on the quality of the geotechnical model, which necessarily requires a proper rock mass characterization.

However, traditional field surveys are particularly complex and unsafe in near-vertical outcrops, with the drawback of collecting information only at the few accessible areas, which are not always representative of the whole study site. Alternative techniques for rock slope investigations were proposed in the last decades with the advent of technologies such as high-resolution digital cameras, laser scanners and drones to overcome the notorious limits of conventional geostructural and geomechanical surveys, highlighted by several authors [15,27,28]. For instance, [29,30] proposed a method to determine joint surface roughness from point clouds obtained by means of Terrestrial Laser Scanning (TLS), [31] introduced a method to monitor landslides by detecting displacement from multi-temporal TLS acquisitions, [32] monitored a Deep-Seated Gravitational Slope Deformation by means of TLS methods and [33] performed rockfall investigation on Digital Elevation Models (DEM) generated from TLS point clouds, with particular reference to kinematically unstable surfaces and block size distribution. In other cases, TLS and Structure-from-Motion (SfM) methods were applied, respectively, for cliff retreat [34] and erosion monitoring [35]. The principles of these technologies, as well as their main advantages, limitations and applications to rock slopes, are illustrated in several articles [36–40].

Nowadays, TLS, Terrestrial Photogrammetry and Unmanned Aerial Vehicle (UAV) techniques are widely used to detect, characterize, model and monitor processes in rock slopes. A very popular issue in the application of remote sensing techniques for rock slope investigations concerns the quantitative characterization of discontinuities from point clouds. As described in [41], discontinuities can be extracted from point clouds by means of semi-automatic or fully automated approaches. The first method makes it possible to calculate the orientation (i.e., dip direction/dip) of discontinuities by calculating the best-fit plane on a geological feature manually marked by the operator in the 3D point cloud. The automatic method requires less interaction and is based on direct segmentation or surface reconstruction. With the direct segmentation method, all the points of the raw dataset distributed along planar surfaces are selected and classified into discontinuities, each one constituted by a mathematical expression. Afterwards, the poles of the main discontinuity sets are identified on stereo plots. The surface reconstruction methods require an initial 3D reconstruction of the point cloud; for example, by means of Triangular Irregular Networks (TINs). All the facets' orientations are plotted on stereonets, allowing the automatic identification of the discontinuity sets. Several authors proposed different algorithms and software solutions for the automatic extraction of fractures from raw or processed point clouds, with all of them being based on the principles outlined above. For instance, some examples of direct segmentation techniques were presented by means of the RANSAC iterative method [42–48]. On the other hand, triangulation techniques were introduced by [28,49–51]. Recently, the semi-automatic or automatic techniques for the extraction of discontinuities in point clouds were optimized with the aim of conducting rock mass classification [52] or identifying additional properties such as discontinuity spacing, persistence, trace length, frequency, intensity, block size and shape [44,53–60]. A detailed review of the methods for structural analyses from point clouds was presented by [61].

The potential of remote sensing techniques for rock slope characterization is widely recognized and includes the ability to acquire data of large areas in reasonably short time periods and under safe conditions, as well as the creation of permanent databases [62–65].

With regard to carbonate systems, in addition to presence of discontinuities and other deformations in the rock mass, they are particularly prone to dissolution, weathering and karstification processes [66–69]. In this case, the interpretation of data from remote sensing techniques might be challenging because of the presence of irregular geometries, especially when dealing with karst or paleo-karst landforms (i.e., weathered materials, karst conduits and cavities), which can affect the accuracy of the automatic methods for rock mass characterization. Due to karst areas being particularly complex and heterogeneous [70,71], specific multidisciplinary on-site investigations are recommended to avoid erroneous geotechnical modelling, estimation of failure susceptibility and planning of intervention strategies [69,72–74], especially in places of high tourist attraction, characterized by higher geological risk due to the presence of people and infrastructure.

Remote sensing techniques in steep carbonate coastal areas are powerful tools to overcome the limits of conventional techniques. However, some practical questions may arise:

1. What is the best technique in terms of costs and benefits for structural analyses and monitoring?
2. Does the type of technology used affect the results of the geostructural characterization and monitoring from point clouds?

In the current literature, few contributions have been presented to answer these questions. For instance, the authors of [34,35], by direct comparison with TLS data, pointed out that the SfM technique can provide results of acceptable accuracy (although not as high as from TLS) in 20% of the time used to collect TLS data. Moreover, [65] compared the stereonets obtained by means of field scanline, TLS and Terrestrial Photogrammetry techniques on rock cuts of different lithologies, while [75] compared the results of TLS and UAV surveys in carbonate environments to illustrate their main advantages and drawbacks. However, to our knowledge, a quantitative and complete comparative analysis of the different remote sensing techniques commonly used for rock mass investigations in coastal areas has not yet been reported.

With this study, we acquired point clouds on a carbonate cliff of high tourist attraction using TLS, Terrestrial Photogrammetry and UAV methods to quantitatively estimate their quality and assess their applicability for rock mass characterization and monitoring by means of semi-automatic techniques. Because of the presence of disturbance elements such as shrubs and human activities, we used two segmentation software applications (Coltop3D [76] and DSE-Discontinuity Set Extractor [77]) rather than triangulation methods to avoid incorrect connection between the points and incorrect surface triangulation [78]. Moreover, the pros and cons of each method are presented in attempt to depict the conditions and goals the different techniques are more suitable for.

## 2. Case Study

The study area is a 20 m high rock cliff facing the Adriatic Sea located at *Lama Monachile* site, in the municipality of Polignano a Mare (southern Italy), in the Apulia region. The site belongs to the eastern part of the Murge area, which is a structural relief of the Apulian carbonate platform formed in Tertiary and overlain by Quaternary deposits [79] (Figure 1). The Apulia region represents the foreland of the Apenninic Chain and has been subjected to tectonic uplift since the middle Pleistocene [80,81]. A series of stepped scarps dipping to NE developed up to the current configuration as a result of the interactions between the tectonic uplift and the absolute sea-level changes [82–84]. The marine terraces are crossed by a karst network of slightly incised valleys, locally named *lame* [85], to which the study site belongs, terminating towards the Adriatic Sea and often difficult to recognize because the waters flow only during exceptional rainfall events, causing flash floods that are typical of karst landscapes [86–88]. The site is made up of whitish to greyish limestones belonging to the Calcare di Bari Formation, unconformably overlain by transgressive calcarenites belonging

to the Calcarenite di Gravina Formation. A pocket beach constituted by loose materials derived mainly from the coastal erosion is located at the base of the cliff in correspondence of the inlet. The rock mass is characterized by sub-vertical joints and thin-to-medium bedded layers whose intersections determine the formation of potential unstable blocks with variable volumes. In addition, notches and karst caves, formed due to wave erosion and dissolution at the interface between fresh groundwater and sea water [89,90], are well visible along the coastline [91]. Mechanical, chemical and biological processes contribute to degradation of the rock mass and influence predisposing instability processes such as toppling, rockfalls, wedge slides and cave failures. Moreover, anthropogenic activities such as artificial cavities in the calcarenites, and terraces and buildings partially carved in the rock mass perturbate the stability conditions of the site, as is also the case for many other towns of Apulia [92]. All these factors represent a serious hazard for the population and infrastructure. Both the old town of Polignano a Mare at the top of the cliff and the beach located at *Lama Monachile* attract many locals and tourists, especially during summer, given the exceptional beauty of the area and the large number of cultural events and sports competitions.

For these reasons, we decided to contribute to the assessment of the stability conditions at the site by means of rock mass characterization and monitoring. Furthermore, we took advantage of the complex geological and geomorphological setting to test different technologies in order to answer to the above questions about the applicability of remote sensing techniques in coastal areas. Different factors, such as scarce accessibility and GPS signal, karst caves, weathered rock materials, Mediterranean vegetation, local regulations and dynamic disturbances (i.e., sea waves and human activities) were of paramount importance to validate the advantages and limitations of remote sensing techniques in complex environments.

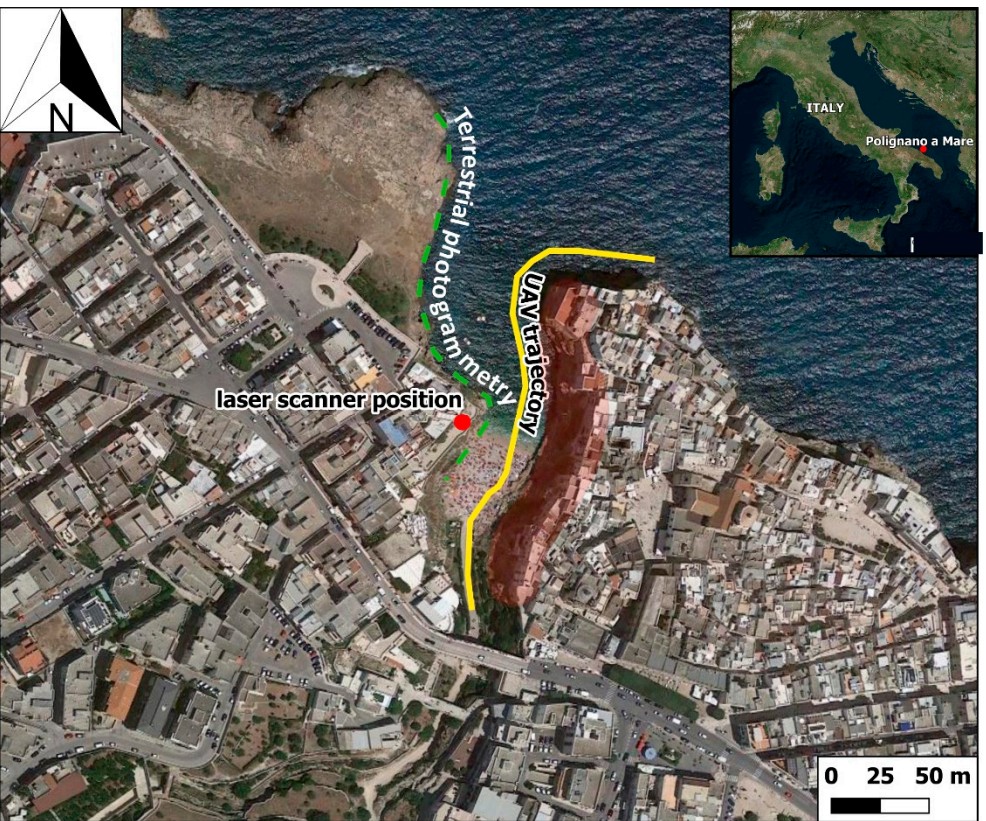

**Figure 1.** Geographic location of the study area (base map retrieved from Google Satellite). The compared area (Figure 2) is highlighted in red.

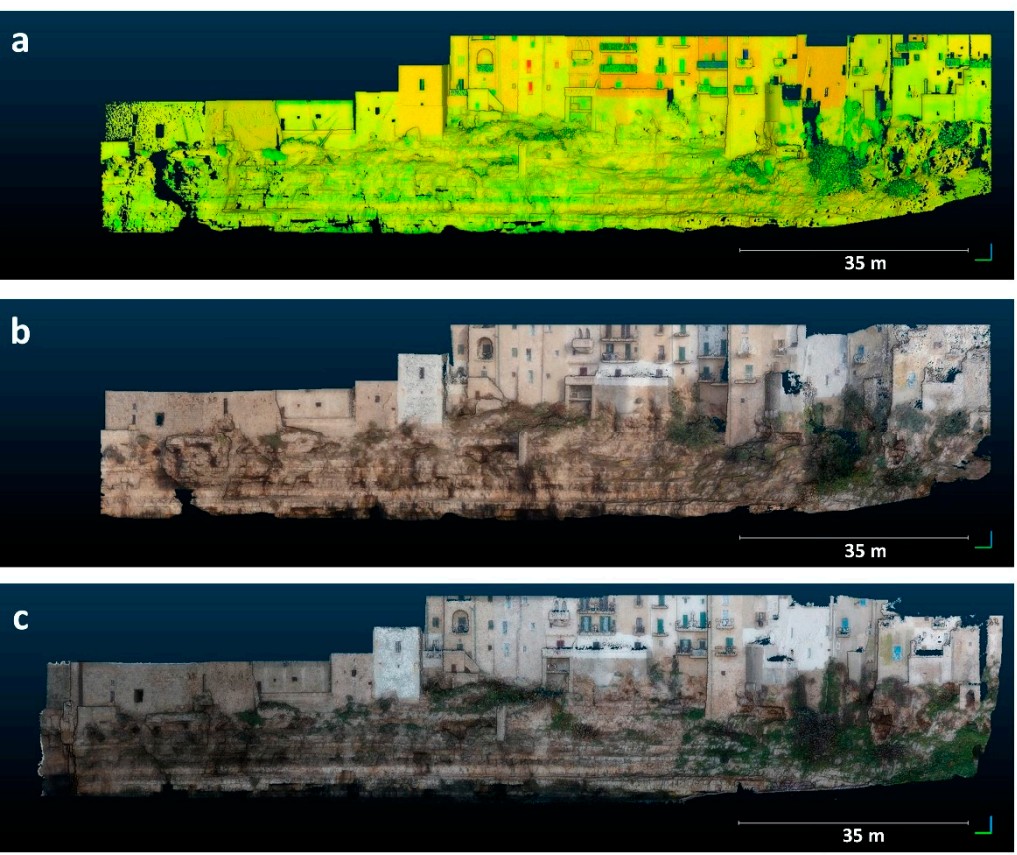

**Figure 2.** Final point clouds acquired by means of remote sensing techniques: (**a**) Terrestrial Laser Scanning point cloud (24,196,954 points and 6852 points/m$^2$ density); (**b**) Terrestrial Photogrammetry point cloud (6,701,071 points and 1507 points/m$^2$ density); (**c**) UAV point cloud (21,849,920 points and 5244 points/m$^2$ density).

## 3. Materials and Methods

### 3.1. Remote Sensing Acquisition and Processing

#### 3.1.1. Terrestrial Laser Scanning

The Terrestrial Laser Scanning point cloud was acquired on 12 December 2019, by means of a Riegl VZ 400 laser scanner, from one scan position located on the opposite side of the cliff (Figure 1), at a 35 m distance. According to the manufacturer's specifications, the laser scanner has a 5 mm accuracy at a range up to 600 m, a measurement rate up to 120,000 points per second and a 360° horizontal field of view [93]. The coordinates of 4 targets placed in different locations of the study side (4 tripods were set on the beach located at the foot of the cliff) were acquired using a Stonex SIII Differential Global Positioning System for accurate georeferentiation of the point cloud. The dataset was imported in CloudCompare software (version 2.12. alpha 2021) and converted from a local to a global WGS84/UTM 33 N metric coordinate system. Successively, points generated by sea reflections and part of the buildings located on the top of the cliff were removed using the segmentation tool. The point cloud was sub-sampled with a 1 cm minimum distance between points to better manage the raw dataset (about 68 million points) on a standard laptop. The resulting point cloud was constituted by more than 24 million points, about 6800 points/m$^2$ density and 1.2 cm mean point spacing (Figure 2a, Table 4).

#### 3.1.2. Structure-from-Motion

Terrestrial SfM Image Acquisition

Terrestrial Photogrammetry was applied on 26 October 2019, using a common digital camera with 4000 × 3000 pixels resolution and 35 mm focal length. One hundred and nine digital photographs with a 1.41 pix/cm mean ground resolution were captured from

different points of view from the opposite side and the base of the cliff, following detailed recommendations [94]. Since the site accessibility was limited, the photos were taken following the trajectory shown in Figure 1, with camera–target distances in the range of 20–100 m. It is outlined that this type of survey was carried out to test the efficacy of low-cost tools, which are easily available on the market, to perform rock slope geostructural analyses and monitoring.

Unmanned Aerial Vehicle (UAV) Image Acquisition

The Unmanned Aerial Vehicle surveys required preliminary planning of the flight mission to achieve optimal coverage of the area, in terms of the function of the morphology of the site, weather conditions and exposition to the sunlight. A manual flight mission with side and frontal overlap, respectively, of 75% and 85% was set for the survey campaign, which was carried out on December 12, 2019. One hundred and thirty frontal photos were acquired using a quadcopter platform DJI Inspire 2, equipped with a 20.8 Megapixel resolution camera, an integrated Global Navigation Satellite System (GNSS) and a remote flight controller, at horizontal distance of 18 m from the cliff (Figure 1). Further details of the UAV system and surveys are reported in Table 1.

**Table 1.** Details of the UAV system, on-board camera and photogrammetric survey.

| UAV SYSTEM | |
|---|---|
| UAV device | DJI Inspire 2 |
| Maximum take-off weight (g) | 4250 g |
| Maximum flight time (min) | 27 |
| Gimbal stabilization | 3-axis (pitch, roll, yaw) |
| ON-BOARD CAMERA PARAMETERS AND SETTING | |
| Camera model | Zenmuse X5S |
| Supported lens | DJI MFT 15 mm 1.7 ASPH |
| Sensor | CMOS, 4/3" Effective Pixels: 20.8 MP |
| FOV | 72° |
| Photo resolution (pix) | 5280 × 3956 |
| SURVEY DETAILS | |
| Flight mode | manual |
| Ground Sampling distance (cm/pix) | 0.41 |
| Coverage area (km$^2$) | 0.00546 |
| Frontal distance from the cliff (m) | 18 |
| Number of photos | 130 |
| Front overlap (%) | 75 |
| Side overlap (%) | 85 |
| Frame shooting interval (s) | 1.5 |
| Number of tie-points | 311,321 |
| Number of projections | 2,290,325 |
| Reprojection error (pix) | 0.541 |
| GCPs XY error (m) | 0.097 |
| GCPs Z error (m) | 0.001 |
| Total GCPs error (m) | 0.010 |

Processing

The images collected by means of Terrestrial Photogrammetry and the UAV systems were processed by means of the Structure-from-Motion (SfM) technique following the workflow of Agisoft Metashape Professional software [95]:

(a) Image inspection, importation, and conversion of the coordinates into the WGS84/33 N metric coordinate system.

(b) Insertion of Ground Control Points (GCP): points whose coordinates were taken from the TLS point cloud on well-recognizable surfaces were added to the photos as constraints to roughly georeference the SfM model. Due to the low resolution,

only 3 GCPs were identified on well-recognizable elements (i.e., building structures) of the Terrestrial Photogrammetry point cloud, whilst 5 GCPs, evenly distributed in the 3-D scene, were detected on the higher-resolution UAV point cloud. The GCP projection errors of the terrestrial and UAV SfM point clouds are, respectively, summarized in Tables 2 and 3. For each GCP, the horizontal ($E_x$, $E_y$) and vertical ($E_z$) reprojection errors correspond to the Root Mean Square Error (*RMSE*) calculated over all the photos where it was visible. The total error for each GCP is given by:
$$RMSE = \sqrt{E_x^2 + E_y^2 + E_z^2}.$$

(c) High-accuracy camera alignment by means of sparse bundle adjustment algorithm [96].
(d) High-quality depth maps calculation and generation of the dense point clouds.
(e) Refinement of the dense point cloud by means of subsampling (minimum distance between points of 1 cm for the UAV point cloud) and direct segmentation.

The final point cloud of the terrestrial SfM consisted of more than 6 million points, with 1507 points/m$^2$ density and 2.5 cm mean point spacing (Figure 2b, Table 4), whilst the point cloud generated by means of UAV SfM was formed by about 22 million points, with a density of 5244 points/m$^2$ and a mean point distance of 1.3 cm (Figure 2c, Table 4).

**Table 2.** Root Mean Square Errors (RMSE) of the Ground Control Points used to georeference the point cloud generated by means of Terrestrial Photogrammetry. The total error in the last row represents the population's standard deviation.

| GCP ID | Number of Images | Horizontal Errors (cm) | | Vertical Errors (cm) | Total Error | |
|---|---|---|---|---|---|---|
| | | X | Y | Z | cm | pix |
| GCPa | 49 | −0.41 | 1.26 | −0.17 | 1.33 | 2.45 |
| GCPb | 55 | 0.27 | −3.09 | 0.69 | 3.18 | 1.15 |
| GCPc | 48 | 0.14 | 1.84 | −0.52 | 1.91 | 0.94 |
| Total | | 0.29 | 2.20 | 0.51 | 2.28 | 1.64 |

**Table 3.** Root Mean Square Errors (RMSE) of the Ground Control Points used to georeference the UAV point cloud.

| GCP ID | Number of Images | Horizontal Errors (cm) | | Vertical Errors (cm) | Total Error | |
|---|---|---|---|---|---|---|
| | | X | Y | Z | cm | pix |
| GCP1 | 18 | 0.76 | −0.96 | 0.15 | 0.12 | 1.27 |
| GCP2 | 29 | 0.00 | 0.63 | 0.04 | 0.63 | 0.68 |
| GCP3 | 57 | 0.04 | 0.10 | 0.03 | 0.11 | 0.40 |
| GCP4 | 29 | −0.44 | 0.63 | 0.15 | 0.79 | 0.24 |
| GCP5 | 49 | −0.35 | 0.70 | −0.51 | 0.94 | 0.23 |
| Total | | 0.42 | 0.80 | 0.29 | 0.95 | 0.56 |

**Table 4.** Summary of the point clouds generated by means of Terrestrial Photogrammetry, TLS and UAV techniques.

| Acquisition Method | Number of Scans | Number of Aligned Photos | Number of Targets/GCPs | Image Pixel Size | Total Reprojection Error of the GCPs | Number of Points in the Point Cloud | Surface Density of the Point Cloud | Average Point Spacing | Number of Points after Sub-Sampling and Cleaning |
|---|---|---|---|---|---|---|---|---|---|
| Terrestrial photogrammetry | / | 109/109 | 3 | 1.41 cm/pix | 2.28 cm | 20,457,182 | 1507 points/m$^2$ | 2.5 cm | 6,701,071 |
| TLS | 1 | / | 5 | / | / | 62,861,985 | 6852 points/m$^2$ | 1.2 cm | 24,196,954 |
| UAV | / | 125/125 | 5 | 0.95 cm/pix | 0.95 cm | 52,363,336 | 5244 points/m$^2$ | 1.3 cm | 21,849,920 |

### 3.2. Quality Assessment of the Point Clouds

Buildings located at the top of the cliff were used to estimate the quality of the point clouds generated with different techniques. Six planes (3 m$^2$ rectangles) were fit in the form of meshes on each point cloud on low-roughness facades of the buildings, evenly distributed in the study area (Figure 3). Then, each point cloud was segmented to obtain six sub-point clouds in correspondence of the planar surfaces. The CloudCompare Cloud-to-Mesh (CtM) algorithm was run to calculate the distances between each sub-point cloud (compared entity) and the corresponding plane (reference entity), and to determine the type of distribution, as well as the mean and standard deviation values. High values of the standard deviation for the normal distribution of the cloud-mesh distances would imply a particular roughness of the point cloud in a clearly flat zone, related to noise, and therefore, would indicate scarce quality.

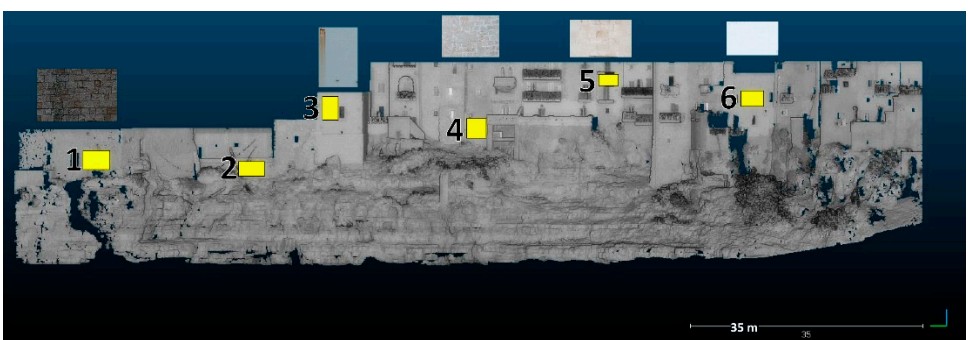

**Figure 3.** Segmentation of 6 rectangles (numbers 1–6) on the facades (photos above) of the buildings for estimating the quality of the three point clouds obtained by means of TLS, terrestrial and UAV photogrammetry.

### 3.3. Comparison of Point Clouds

With the aim of ascertaining the reliability among the different point clouds, and of testing their combined use for a variety of purposes, the acquired point clouds were compared at couples by means of the Cloud-to-Cloud (CtC) distance computation tool available in CloudCompare. To avoid differences caused by different extents of the surveyed area for each acquisition technique, the three original point clouds were segmented together. The TLS and Terrestrial Photogrammetry point clouds were compared, selecting the first as the reference entity and the second as the compared entity. Since the two surveys were not performed on the same day, part of the vegetation was removed by direct segmentation to avoid high values of the scalar field caused by vegetation changes. Further, for the comparison between the TLS and UAV point clouds, that from the laser scanner was kept as a reference. As regards the terrestrial and UAV photogrammetry distance computation, the latter was used as a reference. The default values of the algorithm (i.e., octree level, multi-thread and neighboring points) were used for all the calculations. Visual inspection of the generated scalar field, represented by an appropriate color scale, allowed the identification of zones characterized by higher difference values. More precise details were obtained by overlapping the high values of the scalar field, filtered from the dataset, on the TLS point cloud colored by a grey color scale. In addition, the type of distribution, mean distance and standard deviation of the CtC difference were determined.

### 3.4. Extraction of Discontinuities from Point Clouds

The structural analysis was performed independently on each point cloud using Coltop 3D software [40]. Since the study site is characterized by flat surfaces (terraces) and sub-vertical steps, few planar surfaces could be detected on the point cloud due to their scarce exposition. Thus, a smaller area of the point clouds, where joint sets could be detected as planes (instead of traces), was segmented. As regards the 2-D geostructural analyses of discontinuity traces, which is out of the scope of this research, a specific

methodology was recently proposed in [97]. For each point cloud, after importing the coordinates file, the software automatically computed the point normals and produced a point cloud represented by a Hue Saturation Intensity color scale (HSI) through which the dip direction and the dip of the normals (poles) were represented, respectively, by the hue and saturation values. This graphical representation helps the user to identify the mean discontinuity sets in a point cloud according to their color and saturation. After manually selecting a few polygons that were representative of the main discontinuity sets, Coltop 3D automatically detected the parallel surfaces, with a certain tolerance (in this case, we choose a value of 30°, considered to be appropriate for the study site). The coordinates and dip direction/dip of the points belonging to each discontinuity set were imported in CloudCompare and the respective point clouds were overlapped on the original point cloud (with RGB colors) to validate the results by visual inspection. Dip direction/dip data were then represented on lower-hemisphere Schmidt equal-angle stereographic projections to identify the mean orientation, dispersion and weight of the discontinuity sets.

In a second step, a similar process was carried out using the DSE software [46,54,55]. After the estimation of the normal vectors with 30 nearest neighbors and a 20% tolerance for the coplanarity test, a density-based analysis (number of bins = 64, minimum angle between principal poles = 10°) identified clusters of points with similar orientations, which therefore belonged to the same discontinuity set. Based on previous on-site geostructural characterization and literature data in nearby areas [98], a maximum number of 4 discontinuity sets was set as a threshold. Successively, the mean orientation of each set was calculated and the points were assigned to the respective discontinuity set according to their orientation (setting a 30° maximum angle between the normal vector of a point and of the discontinuity set). Percentages of 20.08%, 13.97% and 20.76% of the points, respectively, remained unclassified for the TLS, photogrammetry and UAV point clouds. The final cluster analysis determined individual discontinuities belonging to each discontinuity set by detecting clusters of points using the DBSCAN algorithm [99]. Therefore, every point belonging to one discontinuity set was assigned to the corresponding cluster. In addition, the persistence and mean spacing for persistent and non-persistent discontinuities were calculated for each discontinuity set using the DSE integrated module.

### 3.5. Rockfall Detection by Means of Multi-Temporal Acquisitions

The accuracy of the TLS and UAV techniques in terms of the detection of potential rockfalls was tested by means of multi-temporal acquisitions with both methods. Two successive surveys were performed using a laser scanner and a drone, in July 2020 and December 2020, respectively. The same operations described in Sections 3.1.1 and 3.1.2 were carried out during the processing phase. Rockfall detection from the two types of datasets was performed according to the method developed in [100,101]. The second point cloud from the TLS survey was roughly aligned on the former by homologous-point pair picking on well-recognizable entities (i.e., anthropogenic elements). The point clouds were segmented together in order to have two datasets of the same area, thus avoiding differences related to missing or additional zones during the comparison.

Successively, a fine registration was applied by means of the Iterative Closest Point (ICP) algorithm [102]. In detail, the original point clouds were split into 6 couples of subsets, and each subset of the second TLS acquisition was finely registered on the corresponding point cloud of the first TLS subset (used as a reference). This operation was iteratively carried out until the roto-translational matrix used to align the compared point cloud on that of the reference became an identity matrix, meaning that the computational limit was reached. The described method made it possible to minimize the Root Mean Square Error of the fine registration (RMSE = 41 mm). Moreover, the TLS point cloud of December 2019 was transformed into a reference triangular mesh using the Poisson Surface Reconstruction plugin [103]. The comparison between the registered and the reference point clouds was carried out using the Cloud-to-Mesh (CtM) distance computation algorithm, which calculates the shortest distance of each point of the cloud from the nearest triangle of

the mesh [93,94]. The calculated scalar value was represented by means of a color scale showing negative surface changes (loss of material) and positive surface changes (gain of material) in blue and red, respectively (Figure 4a).

The same process was used to compare the December 2019 and December 2020 UAV point clouds. To avoid misalignments of the second point cloud, due to insufficient GCPs and "doming deformations" generated from the SfM technique [104–107], the point clouds were segmented in more parts and each subset was registered by means of the ICP algorithm, setting the scaling option. This procedure independently aligns each sub-point cloud on the reference one, and "undeforms" the point cloud. A total RMSE of 79 mm was achieved during this step. The CtM algorithm was run using the mesh of the 2019 point cloud as a reference (Figure 4b).

For both datasets, positive and negative surface changes were compared with digital photographs acquired during the surveys to verify that they were caused by failures of the rock mass and not by vegetation changes or anthropogenic activities. Aiming at detecting source areas of probable rockfalls, greater attention was given to the negative values. The volume of material not detected during the successive acquisition was estimated by merging the reference and compared point clouds, thus isolating the missing rock blocks. At last, the volume was calculated on the mesh obtained by means of the Poisson Surface Reconstruction plugin.

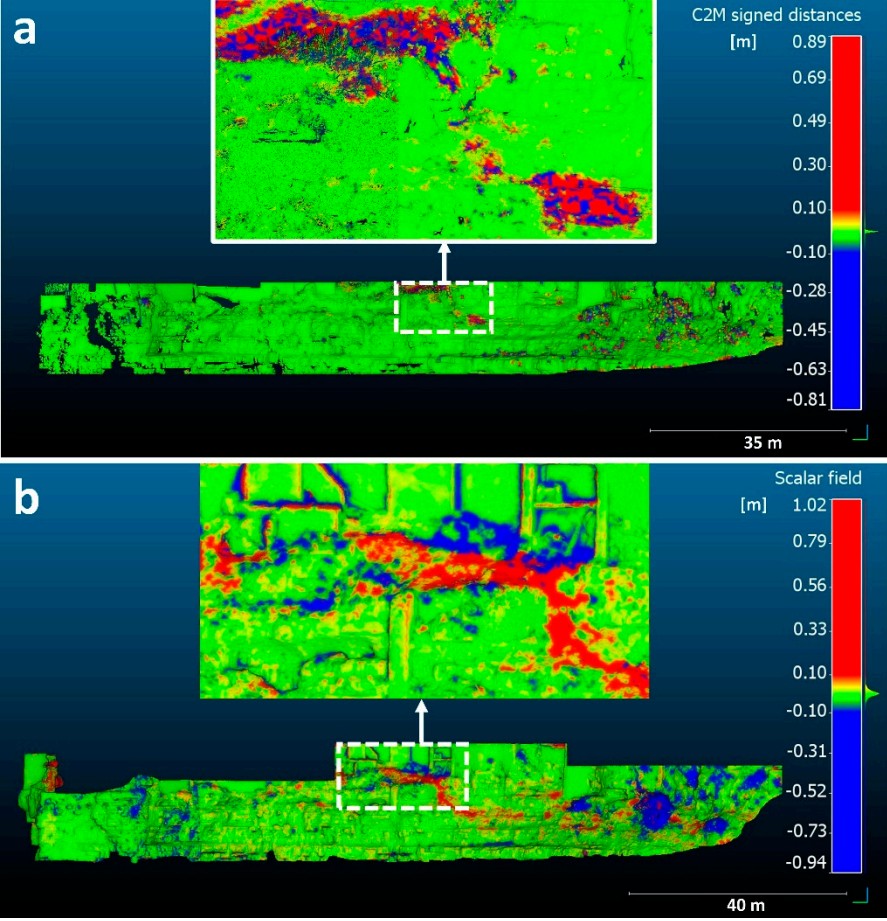

**Figure 4.** Positive and negative surface changes calculated by means of the Cloud-to-Mesh algorithm in consecutive point clouds acquired by different remote sensing techniques. Positive and negative surface changes are colored in red and blue, respectively. (**a**) Differences (in meters) between the December 2019 and July 2020 TLS point clouds; (**b**) differences between the December 2019 and December 2020 UAV point clouds.

## 4. Results and Discussion

### 4.1. Comparison of Point Clouds

As regards the comparison between the TLS and the Terrestrial Photogrammetry point clouds by means of Cloud-to-Cloud distance computation, high values (up to 186 cm) of the scalar field were detected in the southern part of the model (Figure 5). This anomaly was caused by a bad alignment of the compared point clouds, due to an insufficient number of Ground Control Points that could have solved the "doming deformations" generated from the SfM technique [104–107]. Therefore, the ICP algorithm was used to separately align couples of sub-point clouds to optimize the comparison. The mean absolute distance and standard deviation between compared and reference point clouds were 7 and 11 cm, respectively. Ninety percent of the difference values were smaller than 16 cm. Figure 6 shows the results of the CtC distance computation after the fine registration process, with values higher than 10 cm colored in red. Although some misalignment issues were solved, a small portion of the point cloud at the southern sector remained problematic. The red zones that were evenly distributed in the model were related to vegetation changes, human artifacts caused by people moving on the cliff for works during the TLS acquisition, and to TLS occlusions. In fact, the remaining 10% of the distances (with values in the range of 16–186 cm) were detected on elements that were not surveyed from the laser scanner, due to the scan position, located in the areas exposed to NNW and on horizontal surfaces at higher altitudes with respect to the laser scanner.

Although the UAV point cloud misalignments were not consistent, the fine registration was applied to detect the differences related only to the acquisition techniques. The mean and standard deviation values of the CtC differences between the TLS and UAV point clouds were 7 cm and 15 cm, respectively. Since the surveys were carried out at the same time, human activities did not interfere significantly in the differences. For the same reason, vegetation changes were not detected, except for some elements that moved with the wind. Ninety percent of the difference was below 16 cm, while the remaining 10% was up to 186 cm. As shown in Figure 7, the highest values of the differences were detected in areas that were not surveyed by the laser scanner, such as windows and balconies whose surfaces were acquired by the drone flying at higher altitudes with respect to the laser scanner position. Moreover, not all the sub-horizontal surfaces were acquired from the laser scanner, and in some sectors, the high incidence angle caused reflections on the windows located on the buildings in front of the scanner.

To isolate the differences between the point clouds from the reported factors, and to detect only surface changes that occurred due to the reconstruction technique, some cross sections were traced perpendicularly to the previously fitted planes, and the CtC algorithm was run successively. For instance, Figure 8 illustrates a section drawn perpendicularly to plane 6: the mean distance and standard deviations were 2.4 cm and 2.8 cm, while the highest value was 28 cm. Ninety percent of the data were below 5 cm, and higher values were recognized on indented surfaces not acquired by the laser scanner and on the vegetation whose surface was smoothed during the generation of the point cloud by means of the SfM technique.

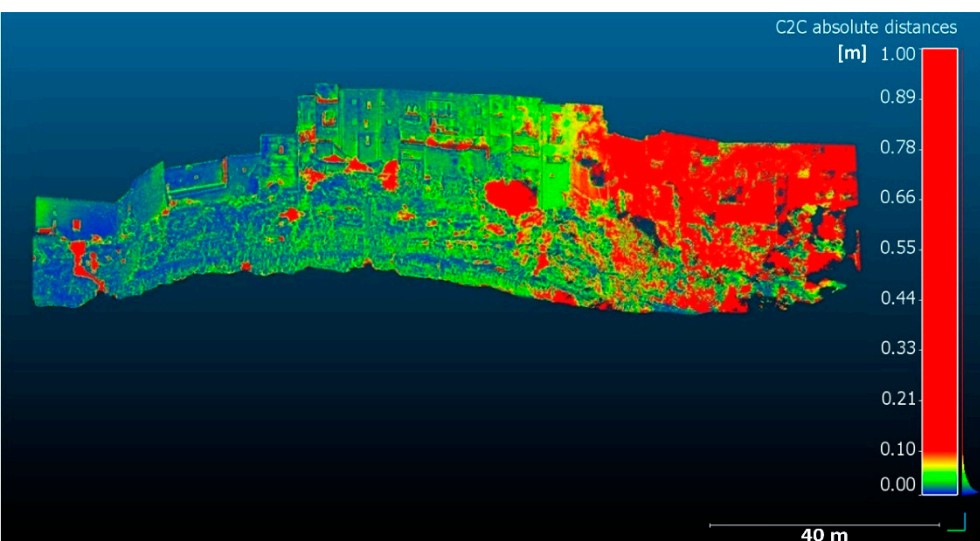

**Figure 5.** Comparison between the point clouds from Terrestrial Photogrammetry (compared) and TLS (reference): Cloud-to-Cloud distance computation in CloudCompare. The greater distances are located in the right part, corresponding to a non-optimal alignment of the compared point cloud due to dome-shaped bias.

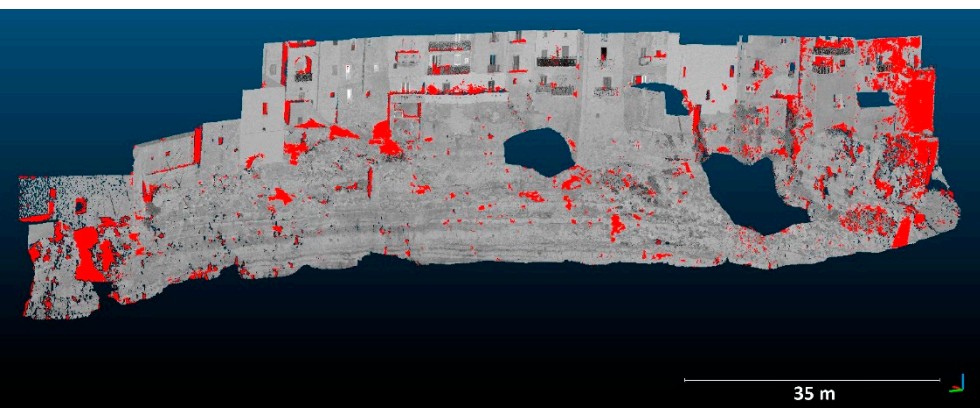

**Figure 6.** Cloud-to-Cloud distance computation between the point clouds from Terrestrial Photogrammetry and TLS (reference) after the fine registration of the first point cloud by means of the Iterative Closest Point (ICP) algorithm. A small part (on the right) of the point cloud obtained by means of photogrammetry was not optimally aligned. Differences higher than 20 cm, corresponding to occlusions, vegetation and human artifacts, are colored in red.

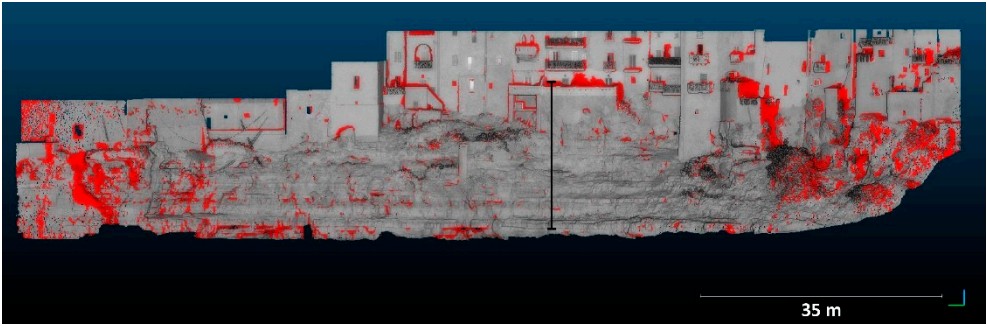

**Figure 7.** Cloud-to-Cloud distance computation between the point clouds from UAV photogrammetry and TLS (reference). The differences higher than 10 cm (red points) were related to occlusions of the TLS technique. The details along the cross section (black segments) are illustrated in Figure 8.

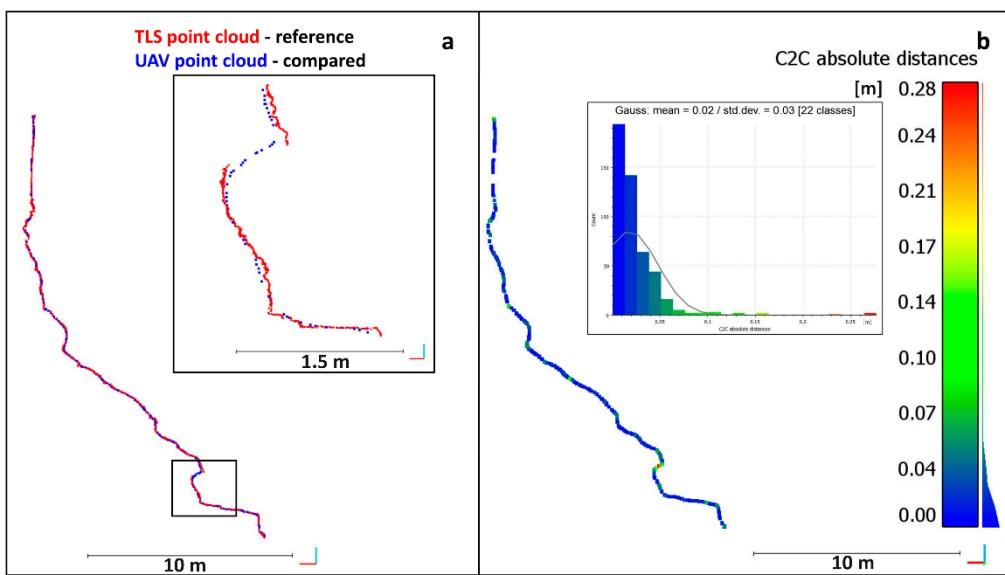

**Figure 8.** (**a**) Cross sections (see Figure 8) of the TLS (reference, in red) and UAV (compared, in blue) point clouds along plane 4. (**b**) A mean distance of 2.4 cm and a standard deviation of 2.8 cm between the two point clouds were detected.

### 4.2. Quality Assessment of the Point Clouds

The scalar values obtained in the distance computation between the sub-sets of the point clouds and the planes fitted on the facades of buildings showed a Gaussian distribution, with peaks on the mean distance (Figure 9). As concerns the Terrestrial Photogrammetry technique, the CtM scalar value in correspondence of plane 3 had a bi-modal distribution, suggesting that the points aligned along two surfaces rather than one. In addition, the measured dip/direction and dip of the six sub-sets of Terrestrial Photogrammetry, TLS, and UAV point clouds along the planes differed by up to one degree (Table 5), implicating that orientation measurements for geometrically simple surfaces (i.e., building facades or walls) are not influenced by the acquisition technique. Although the mean distances of the different types of point clouds along all planes were near to zero, the most significative parameter in terms of usefulness in quantitatively estimating the quality of the datasets was the standard deviation, which indicated the point clouds' dispersion on flat surfaces. Despite similar accuracies of the three techniques being detected for plane 1 and 4, some differences were found for the other datasets. In detail, for each plane except no. 1, the smallest value of the standard deviation of the CtM distances was related to the laser scanning acquisition technique. Slightly higher values, but in the same order of magnitude (mm), were attributable to the UAV technique, while Terrestrial Photogrammetry showed values of up to 4 cm (plane 3). Based on the outcomes of the analysis, it can be deduced that the best quality was provided by the TLS and UAV techniques, followed by Terrestrial Photogrammetry, which might be inappropriate when dealing with complex natural surfaces, especially for in-depth geostructural analyses. However, it must be remarked that terrestrial SfM was performed using a common-low cost digital camera and that the results were quite promising, considering that the photos were taken from different distances because of the complex topography. As a matter of fact, the use of professional tools and longer lenses could have provided higher resolution photos and denser point clouds.

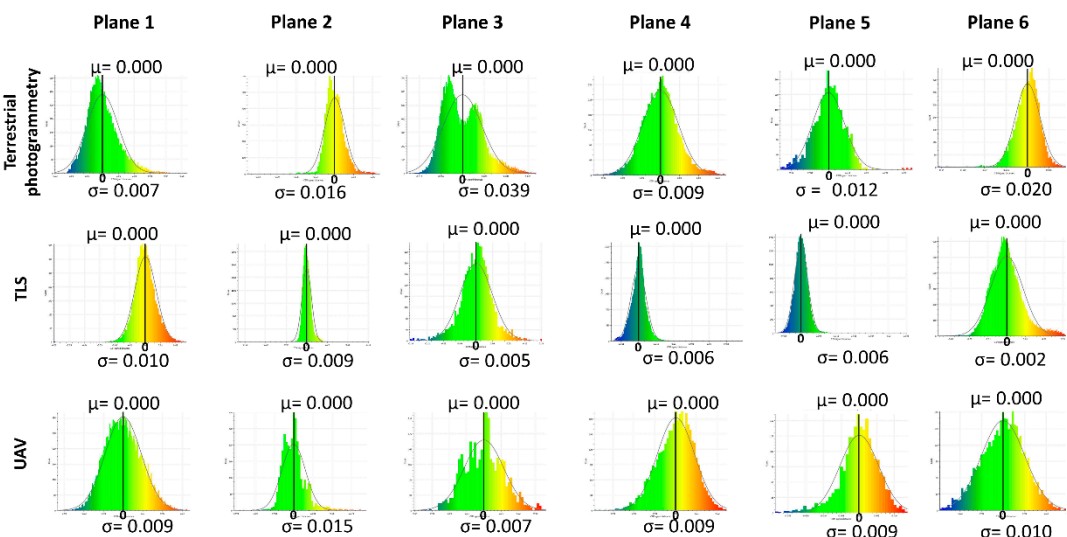

**Figure 9.** Cloud-to-Mesh distance computation on the 6 planar surfaces for the estimation of the point cloud quality obtained from TLS, terrestrial and UAV photogrammetry. μ and σ represent the mean and standard deviation of the distribution of the distances, respectively. High resolution figures are reported in the Supplementary Materials.

**Table 5.** Measured dip direction/dip, mean and standard deviation of the calculated Cloud-to-Mesh distances on the 6 planar surfaces for the estimation of the quality of the point clouds. The mean distances of the subsets of the point clouds from the planar surfaces close to 0 show a good quality for the three datasets. However, higher values of the standard deviation of the cloud from Terrestrial Photogrammetry indicate major irregularity and noise with respect to the other datasets.

| | PLANE 1 | | |
|---|---|---|---|
| Type of Point Cloud | Measured dip direction/dip | Mean CtM distance (m) | Mean st. dev. (m) |
| Terrestrial Photogrammetry | 291/87 | 0.000 | 0.007 |
| Terrestrial Laser Scanning | 291/86 | 0.000 | 0.010 |
| Unmanned Aerial Vehicle | 291/86 | 0.000 | 0.009 |
| | PLANE 2 | | |
| Type of Point Cloud | Measured dip direction/dip | Mean CtM distance (m) | Mean st. dev. (m) |
| Terrestrial Photogrammetry | 78/89 | 0.000 | 0.016 |
| Terrestrial Laser Scanning | 78/88 | 0.000 | 0.009 |
| Unmanned Aerial Vehicle | 78/88 | 0.000 | 0.015 |
| | PLANE 3 | | |
| Type of Point Cloud | Measured dip direction/dip | Mean CtM distance (m) | Mean st. dev. (m) |
| Terrestrial Photogrammetry | 99/88 | 0.000 | 0.039 |
| Terrestrial Laser Scanning | 98/88 | 0.000 | 0.005 |
| Unmanned Aerial Vehicle | 97/87 | 0.000 | 0.011 |
| | PLANE 4 | | |
| Type of Point Cloud | Measured dip direction/dip | Mean CtM distance (m) | Mean st. dev. (m) |
| Terrestrial Photogrammetry | 283/86 | 0.000 | 0.009 |
| Terrestrial Laser Scanning | 283/86 | 0.000 | 0.006 |
| Unmanned Aerial Vehicle | 284/86 | 0.000 | 0.009 |
| | PLANE 5 | | |
| Type of Point Cloud | Measured dip direction/dip | Mean CtM distance (m) | Mean st. dev. (m) |
| Terrestrial Photogrammetry | 104/89 | −0.000 | 0.012 |
| Terrestrial Laser Scanning | 104/88 | −0.000 | 0.006 |
| Unmanned Aerial Vehicle | 104/87 | −0.000 | 0.009 |
| | PLANE 6 | | |
| Type of Point Cloud | Measured dip direction/dip | Mean CtM distance (m) | Mean st. dev. (m) |
| Terrestrial Photogrammetry | 307/84 | −0.000 | 0.020 |
| Terrestrial Laser Scanning | 307/84 | −0.000 | 0.002 |
| Unmanned Aerial Vehicle | 306/84 | 0.000 | 0.010 |

### 4.3. Extraction of Discontinuities from Point Clouds

In agreement with on-site investigations, two joint sets were detected on the Terrestrial Photogrammetry, the TLS and the UAV point clouds using Coltop3D. The best-fit great circles and poles of JS1 and JS2 were reported on equal-angle stereographic projections to determine their mean orientation (Figure 10). Table 6 reports the mean strike, dip, dispersion (Fisher's K parameter) and weight of JS1 and JS2 extracted from the different point clouds, as well as the data collected during conventional geostructural and geomechanical field surveys. With regard to point clouds, the dip direction and dip of the two main joints sets were very similar, with a maximum difference of 4° for the JS2 dip direction measured on the TLS and the photogrammetry point cloud. The Fisher's k parameter was similar for the three datasets, but slightly higher values were obtained from the Terrestrial Photogrammetry. This indicates a minor dispersion of the pole orientation, which might be related to the lower accuracy and density of the point cloud. Similar weight percentages for JS1 and JS2 were detected from the TLS point cloud (49% for JS1 and 51% for JS2, respectively), while the other datasets detected a major weight for JS1 (67% vs. 33% for the Terrestrial Photogrammetry point cloud, and 58% vs. 42% for the UAV point cloud). As stated in the previous section, an underestimation of the weight of JS2 from the laser scanner might have been caused by missing data corresponding to occlusions (due to the position of the laser scanner) along the surfaces exposed to NNW, and thus, with directions parallel to JS2. Field measurements were used to validate the results. The highest orientation difference was found for JS2: a difference of 8° between the field measurements and the Terrestrial Photogrammetry was detected. It is remarkable that the ratios between the weights of JS1 and JS2 for these datasets were inverted. The weight percentages of the joint sets detected on the field were also different from the data extracted from the TLS and UAV point clouds. This discrepancy was attributable to scarce accessibility on the field, which did not allow the sampling of a representative number of surfaces belonging to JS1, leading to its underestimation. However, many fractures belonging to JS2, detected from their traces in the field, were not extracted from the point clouds because of the poor exposition of planar surfaces. In addition, the Fisher's k parameter determined from the field surveys showed a much lower dispersion of the data, probably related to a lower number of measurements, performed on small sectors of the discontinuities. As a matter of fact, field measurements taken by means of a compass-clinometer are representative of small areas and, therefore, they are less affected by undulation and discontinuity planes with respect to remote sensing techniques, which provide measurements of the entire length of the exposed surface.

The DSE software identified two additional discontinuity sets with respect to the previous method, corresponding to the bedding and the ground surface. Since the aim of the analysis was to detect whether the acquisition technique could influence the detection of joint sets, bedding and topography were not considered in the comparison. The main differences between the discontinuity sets extracted using Coltop 3D and DSE are illustrated in Figure 11 and further discussed in Section 5. Tables 7 and 8 summarize the data extracted for JS1 and JS2, respectively, as well as the differences between the pairs of point clouds. A maximum strike difference of approximately 5 degrees was found for JS1 and JS2 in all datasets, but the mean dip direction of JS1, estimated from the UAV point cloud, was opposite with respect to the dip direction extracted from the other point clouds. As regards the dip values, a considerable difference was obtained for JS1: while sub-vertical surfaces were detected from the UAV and TLS point clouds, the photogrammetric process provided less steep values, with a difference of about 16°. The ratios of weight percentages between JS1 and JS2 were similar for the TLS and UAV point clouds, while major differences were found in the dataset from Terrestrial Photogrammetry; however, all detected a greater weight of JS1. Acceptable differences were found for the persistence estimation for JS2, but not for JS1: the maximum persistence estimated from the Terrestrial Photogrammetry point cloud differed by about 3 m from the other datasets. A possible explanation for this difference is that more discontinuities were grouped together into a single cluster because

of the lower resolution of the Terrestrial Photogrammetry point cloud, thus leading to an overestimation of the persistence (see blue discontinuities in Figure 11a).

With regard to spacing, considering both persistent and non-persistent joints as described by [54], the results of the UAV and TLS point clouds (which are similar) were much lower than those calculated from the Terrestrial Photogrammetry technique, which in some cases showed values that were 10 times higher.

Based on the outcomes of the two approaches to detect and characterize the discontinuity sets from point clouds, it was found that the most accurate results were detected from the UAV point cloud, which made it possible to overcome the limitations of the TLS technique (occlusions could not have been avoided because of scarce accessibility at the study site) and of the geostructural and geomechanical surveys. Instead, an inappropriate characterization resulted from the terrestrial photogrammetric survey. The mean orientations of the discontinuity sets extracted from the three point clouds by means of Coltop3D and DSE, together with the results of field surveys, are reported in Table 9.

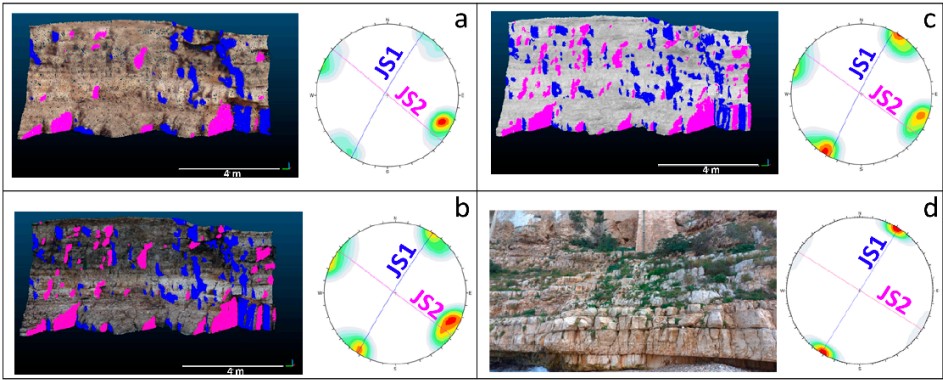

**Figure 10.** Semi-automatic identification of the main joint sets from the point clouds in a representative area of the rock mass with Coltop 3D (**a**–**c**) and validation by means of field surveys (**d**). (**a**) Point cloud and stereonet from Terrestrial Photogrammetry; (**b**) point cloud and stereonet from Unmanned Aerial Vehicle photogrammetry; (**c**) point cloud and stereonet from Terrestrial Laser Scanning; (**d**) photo and stereonet from field surveys. The stereonets show the best-fit great circles and poles of the main joint sets, and high lower-hemisphere Schmidt equal-angle stereographic projections (Figure S2).

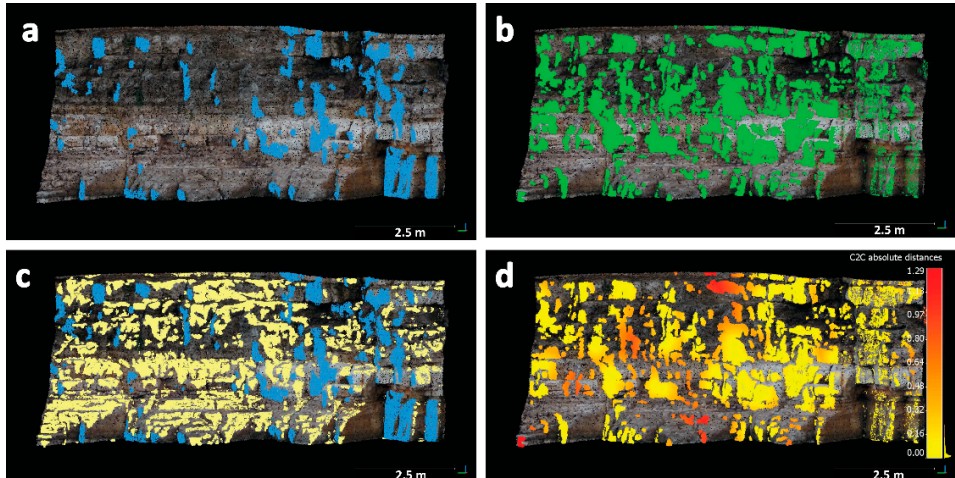

**Figure 11.** JS1 extracted from Coltop (**a**) and DSE (**b**). (**c**) Some points of the topography (in yellow) were not separated from JS1 (blue) by DSE and interfered with the estimation of the JS1 dip direction. (**d**) Differences in J1 between the point clouds from Coltop and DSE: red areas correspond to points belonging to the topography that were considered by the DSE for the extraction of JS1.

**Table 6.** Main orientation, Fisher's K parameter and weight of the main discontinuity sets detected from the Terrestrial Photogrammetry, TLS and UAV techniques by means of Coltop3D, and results of the field surveys.

| Acquisition Technique | JS1 | | | | JS2 | | | |
|---|---|---|---|---|---|---|---|---|
| | Dip Direction | Dip | Fisher's K | Weight % | Dip Direction | Dip | Fisher's K | Weight % |
| TLS (21,828 poles) | 301 | 86 | 40 | 48.55 | 216 | 90 | 41 | 51.45 |
| Photogrammetry (8690 poles) | 300 | 85 | 42 | 66.60 | 40 | 89 | 48 | 33.40 |
| UAV (16,697 poles) | 302 | 86 | 37 | 57.62 | 217 | 90 | 44 | 42.38 |
| Field measurements (216 poles) | 301 | 90 | 59 | 33.93 | 212 | 90 | 216 | 66.07 |

**Table 7.** Characterization of JS1 by means of Discontinuity Set Extractor (DSE) on point clouds acquired from Terrestrial Photogrammetry, TLS and UAV (top) and calculated differences (bottom).

| Data from geostructural analysis | | | | | | | | |
|---|---|---|---|---|---|---|---|---|
| Technique | Dip dir. ° | Dip ° | Density | % | Persistence (m) | | Spacing (m) | |
| | | | | | mean | max | persistent | non-persistent |
| TLS (24,779 poles) | 282.73 | 87.14 | 0.70 | 30.86 | 0.34 | 2.05 | 0.14 | 0.04 |
| Photogrammetry (4784 poles) | 283.28 | 70.69 | 1.79 | 39.87 | 0.68 | 5.37 | 0.62 | 0.40 |
| UAV (17,123 poles) | 108.17 | 86.84 | 0.37 | 26.59 | 0.42 | 2.61 | 0.19 | 0.06 |
| Differences | | | | | | | | |
| Compared dataset | Dip dir. ° | Dip ° | Density | % | Persistence (m) | | Spacing (m) | |
| | | | | | mean | max | persistent | non-persistent |
| TLS-photogrammetry | 0.55 | 16.45 | 1.09 | 9.01 | 0.35 | 3.32 | 0.48 | 0.36 |
| TLS-UAV | 174.56 | 0.30 | 0.34 | 4.27 | 0.08 | 0.56 | 0.05 | 0.02 |
| UAV-photogrammetry | 175.11 | 16.15 | 1.42 | 13.28 | 0.27 | 2.77 | 0.44 | 0.34 |

**Table 8.** Characterization of JS2 by means of Discontinuity Set Extractor (DSE) on point clouds acquired from Terrestrial Photogrammetry, TLS and UAV (top) and calculated differences (bottom). The persistence and mean spacing are calculated over the area shown in Figure 10.

| Data from geostructural analysis | | | | | | | | |
|---|---|---|---|---|---|---|---|---|
| Technique | Dip dir. ° | Dip ° | Density | % | Persistence (m) | | Mean spacing (m) | |
| | | | | | mean | max | persistent | non-persistent |
| TLS (16380 poles) | 31.18 | 87.96 | 1.08 | 20.40 | 0.34 | 2.02 | 0.20 | 0.08 |
| Photogrammetry (578 poles) | 35.24 | 86.97 | 0.42 | 6.96 | 0.50 | 1.64 | 0.62 | 0.40 |
| UAV (9325 poles) | 29.35 | 90.00 | 0.48 | 14.48 | 0.40 | 1.92 | 0.27 | 0.13 |
| Differences | | | | | | | | |
| Compared dataset | Dip dir. ° | Dip ° | Density | % | Persistence (m) | | Mean spacing (m) | |
| | | | | | mean | max | persistent | non-persistent |
| TLS-photogrammetry | 4.06 | 0.99 | 0.66 | 13.44 | 0.16 | 0.38 | 0.42 | 0.32 |
| TLS-UAV | 1.83 | 2.04 | 0.60 | 5.92 | 0.06 | 0.10 | 0.07 | 0.05 |
| UAV-photogrammetry | 5.89 | 3.03 | 0.07 | 7.52 | 0.10 | 0.28 | 0.35 | 0.27 |

**Table 9.** Orientation of the discontinuity sets extracted by means of Coltop3D and DSE on the three datasets.

| | JS1 | | | | JS2 | | | |
|---|---|---|---|---|---|---|---|---|
| | Dip direction | | Dip | | Dip direction | | Dip | |
| | COLTOP3D | DSE | COLTOP3D | DSE | COLTOP3D | DSE | COLTOP3D | DSE |
| TLS | 301 | 283 | 86 | 87 | 216 | 31 | 90 | 88 |
| Photogrammetry | 300 | 283 | 85 | 71 | 40 | 35 | 89 | 87 |
| UAV | 302 | 108 | 86 | 87 | 217 | 29 | 90 | 90 |
| Field measurements | 301 | | 90 | | 212 | | 90 | |

### 4.4. Rockfall Detection by Means of Multi-Temporal Acquisitions

Positive and negative surface changes between pairs of point clouds acquired using the same technique in different periods were analyzed in detail.

All positive changes, both for the UAV and TLS pairs of point clouds, were related to the growth of vegetation after the first acquisition (Figure 12). It is specified that an initial attempt to semi-automatically filter the vegetation in both point clouds was made by means of the Canupo plugin [108]. However, shrubs and bushes grown in the fractures and cavities of the rock mass were not correctly identified because of their similarity in shape and color (with respect to the darker lithofacies) to the rocks. As consequence, not all the vegetation was segmented out of the point clouds and some areas of the rock slope were removed as well (especially in indented areas), causing several surface changes to be detected by the Cloud-to-Mesh distance computation. For this reason, the vegetation was kept during the point cloud comparison process and identified by means of visual inspection during the interpretation phase.

The negative surface changes in the TLS point clouds were more difficult to interpret: although some were clearly determined by human activity and changes in the vegetation cover (Figure 13a), the low resolution of the photos taken by the laser scanner during the acquisitions did not make it possible to distinguish whether the points missing in the successive point cloud were related to the loss of material or vegetation, which, in some cases, is very similar in color and shape to the darker lithofacies. On the contrary, interpretation of the negative deviations was much accurate for the UAV point clouds: loss of vegetation and of man-made elements could be detected by comparing the high-resolution photos (Figure 13b–d). However, the fine registration of the successive UAV point cloud to the reference one was not as precise as the one of the TLS point clouds because of the SfM limitations. In fact, due to the impossibility of acquiring photos during the second flight from the same position and perspective of the first flight, not all the points could be matched during the ICP registration. As a consequence, more surface changes were observed with respect to the laser scanner point clouds, and more time was necessary for the interpretation. In addition, the detection threshold (4 cm) for surface changes was higher than the TLS one (1.5 cm), meaning that only changes higher than 4 cm could be detected from the UAV point cloud. Only one probable rockfall was identified from the two datasets (Figure 14). The estimated volumes were 0.038 m$^3$ and 0.066 m$^3$ for the TLS and UAV point clouds, respectively. A Cloud (UAV) to Mesh (TLS) distance computation of the point clouds acquired on the same day in December 2019 from the drone and from the laser scanner allowed the detection of the main differences (Figure 15). The positive surface changes (red points in Figure 15a) were located on the indented surface of the rock block (which failed 6 months later) because of inaccurate reconstruction of the point cloud from the SfM technique. Smoother surfaces caused the generation of a larger mesh by means of Poisson Surface Reconstruction with consequent overestimation of the failed rock block volume (Figure 15b,c). Although an anthropogenic cause of the failure cannot be excluded, the applied method shows that the TLS data were more accurate in terms of the estimation of the rockfall volume.

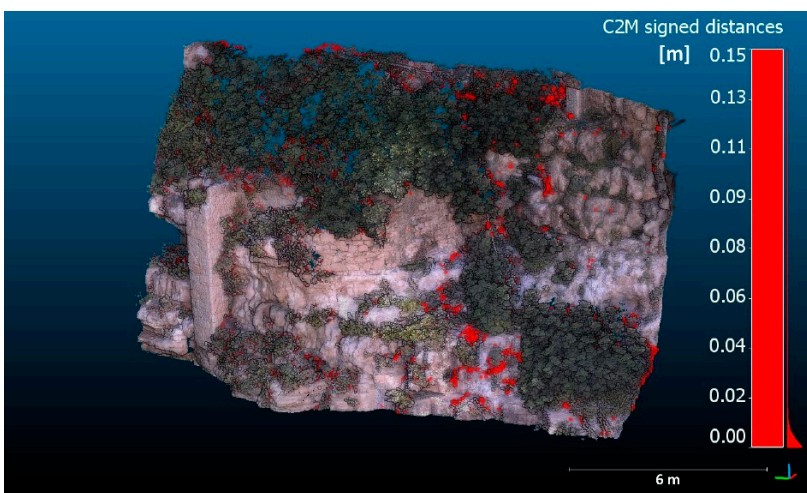

**Figure 12.** Comparison between the point clouds obtained by means of the TLS technique in December 2019 and July 2020: the positive deviations correspond to the growth of vegetation after the first acquisition.

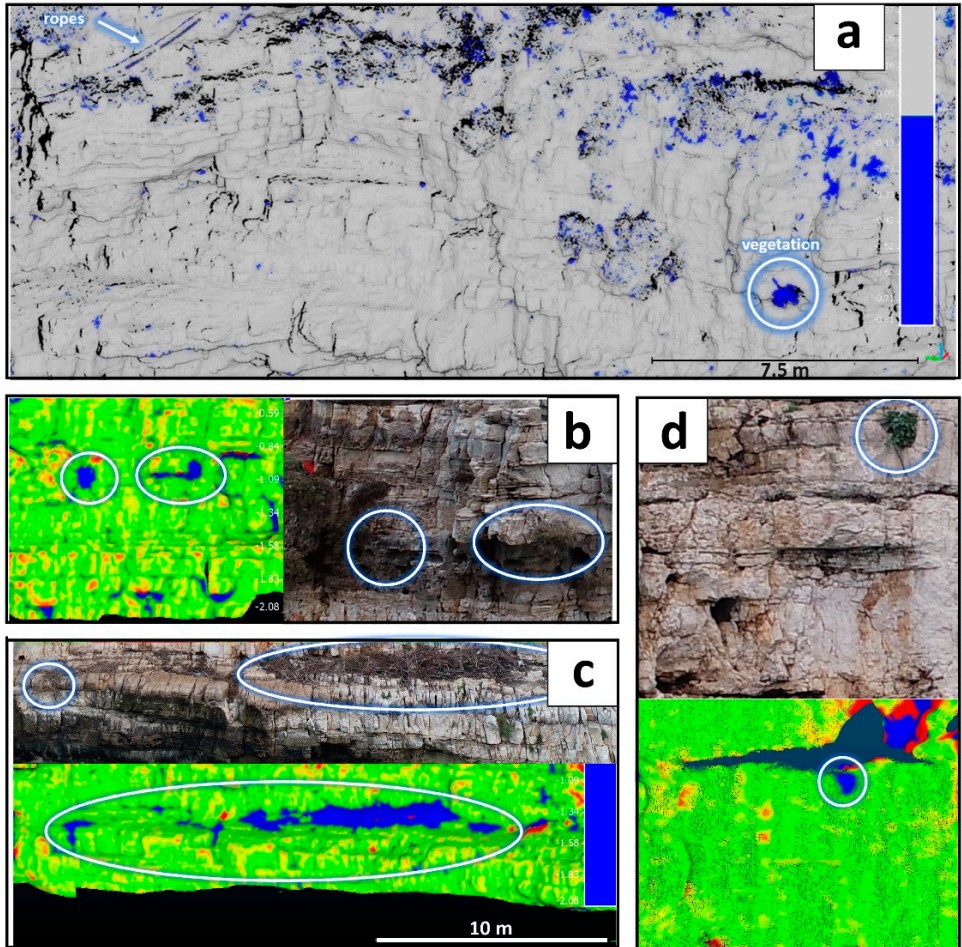

**Figure 13.** Negative surface changes detected in the TLS and the UAV point clouds. (**a**) The negative deviations of the TLS point clouds were mostly related to the loss of vegetation and removal of human artifacts (i.e., ropes) after the first acquisition. (**b–d**) The negative deviations of the UAV point clouds were identified directly on the high-resolution photos acquired by the drone and correlated to the loss of vegetation, whose color and shape could be easily confused with portions of the rock mass in the point clouds.

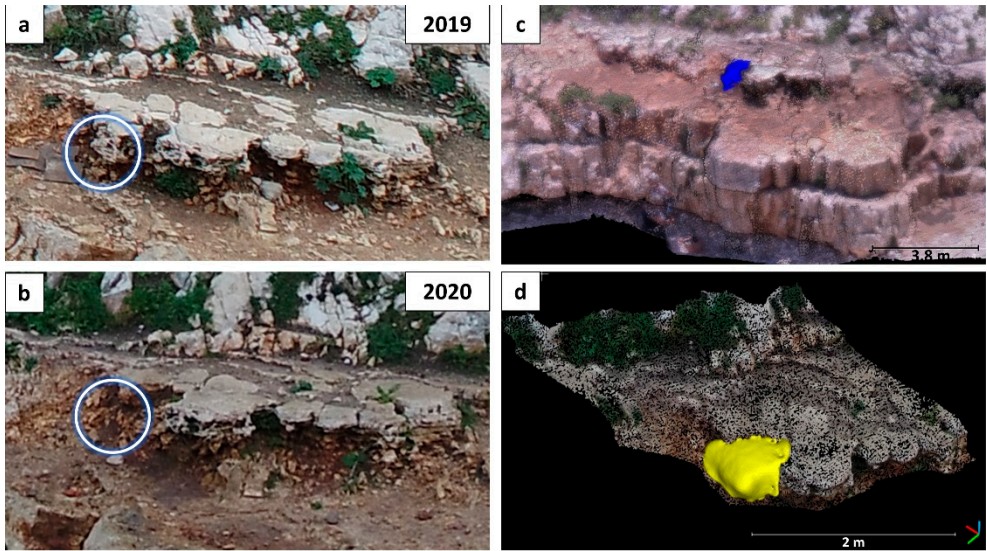

**Figure 14.** Rockfall occurred between the 2019 (**a**) and the 2020 (**b**) acquisitions, detected in the form of negative deviation from the TLS (**c**) and UAV (**d**) point clouds.

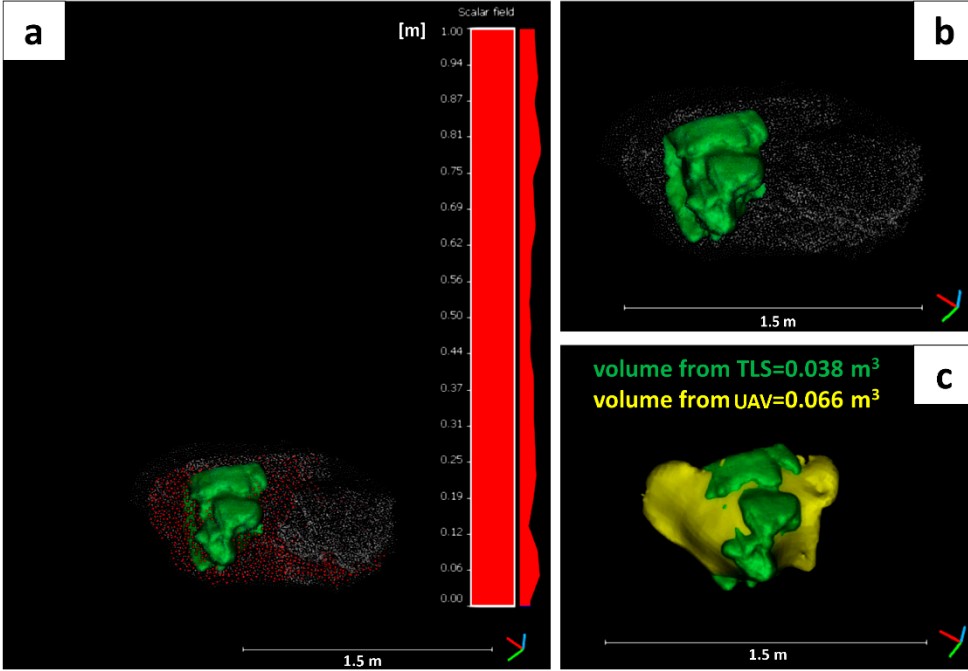

**Figure 15.** Details of the rockfall detected in the point clouds from the UAV and TLS acquisitions' comparisons. (**a**) Cloud (UAV) to Mesh (TLS) distance computation of the point clouds acquired on the same day in December 2019, with the positive deviations highlighted in red; (**b**) mesh of the rockfall detected in the TLS point clouds; (**c**) comparison between the volumes of the meshes calculated using the TLS (green) and UAV (yellow) techniques. The higher volume of the rockfall, as detected from the UAV technique, was due to a limitation of the SfM technique, which was not able to produce a detailed reconstruction of the surface in the fissures of the rock mass. Therefore, the smoother surface led to an overestimation of the volume of the detected rockfall.

## 5. Main Outcomes and Conclusions

Remote sensing techniques are of paramount importance to overcome the limitations of conventional field methods for rock slope investigations, especially when dealing with poorly accessible areas of large size and with unsafe conditions. Since several technologies have been introduced in recent decades, the question as to which method is more appropri-

ate for a particular case study may arise. It is evident that the choice of technique depends on the geologic, morphological and environmental conditions of the target, as outlined by [109]. With this study, we performed a comparative analysis of point clouds obtained by means of TLS, terrestrial and UAV photogrammetry, after validation by means of conventional geostructural and geomechanical surveys, in order to evaluate the advantages and limitations of each technique for slope investigations in complex coastal areas, which are of high interest in terms of the tourist economy.

The TLS methods allowed the acquisition of detailed point clouds in a relatively short time at distances of up to hundreds of meters, depending on the laser scanner used. Conversely, the UAV techniques required a preliminary mission-planning phase to achieve the best coverage area with the most appropriate ground resolution. Furthermore, permission from the authorities may be mandatory in flight-restricted areas. Adverse weather conditions such as rain, wind or fog do not affect laser scanner acquisitions as much as terrestrial and UAV photogrammetric surveys, for which the quality of the photos might be seriously compromised, in addition to the risk of instrument damage. For photogrammetry, partially cloudy weather is preferred to sunny conditions [106] to avoid shadows and sea reflections, which can cause uneven textures and areas with low density or artifacts in the processed point cloud. However, UAV techniques require less effort in terms of costs and logistics, which can be an issue for TLS methods in poorly accessible and steep sites, due to difficulties in carrying the laser scanner and targets. TLS can be inadequate for vertical or sub-vertical slopes if the scan positions are limited by the site morphology. As a matter of fact, occlusions along the bedding and surfaces exposed to NNW in the case study were caused by the impossibility of higher positioning of the laser scanner, and occurred further north with respect to the scan position. However, as stated by several authors [73], TLS techniques are generally able to acquire data in vegetated areas, depending on the type of vegetation, which can be an issue for SfM surveys. Terrestrial Photogrammetry is not the best solution in poorly accessible areas as well; for instance, a sector of the Terrestrial Photogrammetry point cloud (right part of Figure 9) appeared distorted when compared to the TLS dataset because of the lack of good camera locations and the consequent misalignment during the SfM procedure. In addition, if the camera positions are constrained by terrain morphology, a change of the distance from the outcrop can cause differences in the ground resolutions in the 3D model.

As regards the processing phase, photogrammetry techniques are more time-consuming with respect to the TLS method, because more steps are necessary to carry out the SfM procedure, which can take some hours of work (from image inspection to software computation) depending on the number of photos to process and the desired resolution. In addition, a lack of sufficient Ground Control Points or a low GPS signal of the UAV system may cause deformations and incorrect georeferentiation, with unreliable results consequently being obtained in the geostructural analyses, especially if the sectors of the point clouds are mutually shifted.

Moreover, the cleaning phase of the photogrammetric technique requires greater efforts from the operator because of numerous unwanted objects such as the background and dynamic disturbances (i.e., sea waves) affecting the point cloud of the presented case study. It is remarkable that such elements had to be manually segmented out because the classification algorithms available in Agisoft Metashape and CloudCompare also removed points belonging to the rock mass in sectors with similar colors. On the other hand, the TLS point cloud appeared much less disturbed from reflections and scatter points, but in some cases, dynamic disturbances affected the quality of the point cloud. Specifically, the TLS point cloud was locally affected by noisy points caused by people moving on the study area; the same obstacles were easily removed when implementing the SfM technique by applying masks. Based on these considerations, we specify that, although SfM techniques can reduce the data collection times by about 80% [35], the same amount of time is spent in producing and cleaning the final point cloud. However, if the surveyed area is affected by

dynamic disturbances, SfM techniques may provide a better solution in terms of accuracy and time.

As shown in Figure 9 and Table 5, the TLS and UAV techniques performed at short distance from the target provided point clouds with similar accuracies: for the less accurate zones (i.e., plane 1 for TLS and plane 2 for UAV), 1.3% of the points were at 3 and 4.5 cm from the reference plane ($\mu \pm 3\sigma$). In non-occluded areas, the TLS point cloud was able to provide details of very small features (i.e., building elements) and of zones below the vegetation, which were not observable from the UAV dataset. Conversely, for plane 3 of the Terrestrial Photogrammetry point cloud (Table 5), 1.3% of the points were at a distance of 12 cm from the reference plane ($\mu \pm 3\sigma$). Although the same plane orientations were estimated from the three datasets, the low quality of the Terrestrial Photogrammetry point cloud might represent a limitation when dealing with complex morphologies.

With regard to the geostructural analysis, it was found that the extracted main discontinuity sets had almost the same orientation and that only their weight was differently estimated because of occlusions specific to the analyzed dataset. The similarity of the results from geostructural characterization from TLS and Terrestrial Photogrammetry along rock cuts was reported in [63]. The discontinuities were sub-vertical; therefore, the estimated dip direction could be towards one direction or the opposite one. The variation of polarity for steep discontinuities, related to undulated patterns, was also outlined in [73]. In summary, the main differences of the geostructural analysis do not depend on the point cloud type, but on the approach used. Specifically, a difference of almost 20° was found for the J1 orientation from Coltop3D and DSE. Through a detailed analysis it was assessed that the DSE did not manage to separate JS1 from the ground surface; therefore, some points belonging to the mean orientation of the cliff (N–S) were merged with the DS and its mean strike was more oriented towards N–S with respect to the results of Coltop3D (JS1 was properly isolated in Coltop3D by means of the operator's validation). In other words, some secondary structures generated from discontinuities of larger scale belonging to JS1 and JS2 were attributed to a third discontinuity set from the DSE, which was not accurately separated from JS1 (Figures 11 and 16). Based on these observations, it is believed that automatic methods should be performed on UAV point clouds rather than on TLS point clouds in order to easily validate the results by means of visual inspection of more detailed textures or high-resolution photos.

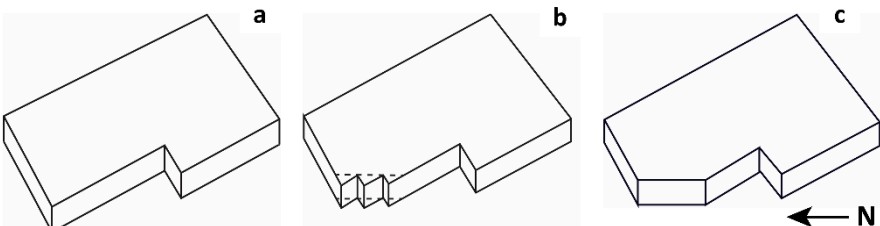

**Figure 16.** Large-scale orthogonal discontinuities (**b**) developed in the bedding (**a**) were eroded over time and led to the formation of sub-vertical discontinuities striking N–S (**c**) that were not correctly separated from the joint set JS1 by DSE software. For this reason, the mean dip direction of JS1, calculated by means of DSE, was about 20° closer to the North with respect to that calculated by means of Coltop3D software.

The results of multi-temporal acquisitions obtained by means of the TLS and UAV techniques showed that major accuracy, in terms of monitoring, can be achieved from the first one, given the lower threshold limit to detect surface changes. Indeed, the TLS instrumental errors are small enough to ensure the detection of millimetric deformations when comparing consecutive point clouds [110]. Interpretation of the UAV results was more complex because of the presence of more anomalies, which were due to differences in shadow zones depending on the trajectory of the drone during the two acquisitions. However, the origin of the surface change was less uncertain with respect to the TLS

method because the vegetation was easily recognizable from the high-resolution photos. In addition, UAV systems allow the monitoring of larger zones compared to the laser scanner, whose field of view can be limited in poorly accessible zones. The main issues involved in using UAV techniques for rockfall detection and cliff monitoring are related to the poor coverage of the SfM in the indented areas, which, in this study, caused an overestimation of the failed block volume. This limitation should be taken into account when planning protection measures such as nets.

Based on the main outcomes of the comparative analysis, Terrestrial Photogrammetry can be used to perform only preliminary slope investigations in complex environments. However, professional cameras with longer focal lengths could be used to obtain higher-resolution photos and larger ground resolution. More detailed geostructural analyses and monitoring can be obtained using both the TLS and UAV methods. Each technique has its own advantages and drawbacks, but the authors agree with the statements made in [109], where it was stated that accessibility is the first factor to consider in order to choose the most appropriate technique. If accessibility is ensured, the most suitable technique is TLS, in terms of both accuracy and time.

Being complementary to each other, both methods can be combined to fully characterize the examined area. Specifically, TLS surveys can be carried out to generate accurate and geo-referenced point clouds to use as references. Successive UAV acquisitions are recommended to increase the observation area with less effort, collect data in the occlusions of the TLS point clouds (if present) and detect large-scale surface changes. If surface changes are detected, specific TLS surveys can be applied to the instable area to correctly estimate the volume of the failed blocks.

Finally, we remark that remote sensing techniques are powerful tools for rock slope characterization and help to significantly reduce the time needed to carry out conventional surveys in safe conditions. However, field surveys remain irreplaceable in the collection of information on the mechanical behavior of rock masses and should be combined with the described technologies to achieve deep knowledge of the investigated area.

**Supplementary Materials:** The following are available online at https://www.mdpi.com/article/10.3390/rs13245045/s1, Figure S1: Cloud-to-Mesh distance computation on plane 1–6 for the estimation of the point cloud quality obtained from TLS, terrestrial and UAV photogrammetry. (μ and σ represent the mean and standard deviation of the distribution of the distances, respectively). Figure S2: Semi-automatic identification of the main joint sets from the point clouds in a representative area of the rock mass with Coltop 3D (**a–c**) and validation by means of field surveys (**d**), high lower-hemisphere Schmidt equal-angle stereographic projections.

**Author Contributions:** Conceptualization, L.L. and M.J.; Validation, G.F.A. and M.-H.D., Formal Analysis, L.L.; Investigation, L.L., G.F.A. and M.J.; Resources, M.P. and M.J.; Data Curation, L.L. and M.-H.D.; Writing—Original Draft Preparation, L.L.; Writing—Review and Editing, G.F.A., M.-H.D. and M.P.; Visualization, L.L. and M.-H.D.; Supervision, G.F.A., M.-H.D. and M.P.; Funding Acquisition, M.P. All authors have read and agreed to the published version of the manuscript.

**Funding:** This research received no external funding.

**Data Availability Statement:** The datasets generated and analyzed during the current study are available from the corresponding author on reasonable request.

**Acknowledgments:** We thank Trimble Geospatial company (www.trimble-geospatial.it—accessed on 20 November 2021) for providing the TLS point cloud of July 2020 used to analyze multi-temporal acquisitions for rockfall detection. We also thank Antonella Marsico and Marco La Salandra from the University of Bari, respectively, for the 2019 TLS survey and for the UAV mission planning and surveys. A special thanks goes to the Reviewers who, with their observations, have made it possible to significantly improve this work.

**Conflicts of Interest:** The authors declare no conflict of interest.

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
