# Peer review of "Comparison of Remote Sensing Techniques for Geostructural Analysis and Cliff Monitoring in Coastal Areas of High Tourist Attraction: The Case Study of Polignano a Mare (Southern Italy)"

_remotesensing, doi:10.3390/rs13245045_

Round 1

Reviewer 1 Report

The manuscript presents a very accurate characterization of a coastal cliff of a site with high tourist attraction.
The critical analysis of the techniques commonly used in the literature is exhaustive and discussed in detail. The methodology adopted for the study of the study site is presented with extreme accuracy, highlighting the positive and negative aspects of each single technique. The description of the results is clear and perfectly congruent with the data presented. The conclusions also provide interesting considerations on the validity and applicability of each methodology and above all highlight the importance of the control of field surveys for the evaluation of the mechanical behavior of rock masses.
The paper is well organized, written in a language that is always understandable; assumptions are always supported by data analysis; the purpose of the work is clear and well illustrated.
Based on the considerations made, the manuscript can be accepted without revisions.

Author Response

Dear Reviewer,

thank you very much for your positive feedback.

Sincerely,

the authors

Reviewer 2 Report

Review of the manuscript “Comparison of remote sensing techniques for geostructural analysis and cliff monitoring in coastal areas of high tourist attraction: the case study of Polignano a Mare (southern Italy)” by Loiotine and coauthors, submitted as a technical note to Remote Sensing.

The paper presents an interesting and useful comparison of three techniques commonly applied for remote surveys of rock slopes rock slopes, i.e., digital photogrammetry, Terrestrial Laser Scanning and Unmanned Aerial Vehicle. The rock cliff is of limestone and dolostone, which, being subject to surface dissolution, make automatic fracture detection particularly challenging. The manuscript is accurately presented in both text and figures. The quantitative comparisons of the output from the three methods are sound and well discussed. I therefore recommend publications, although some minor issues remain to be clarified/fixed as detailed in my comments below.

Best regards

Silvia Mittempergher

General comments

  • The terrestrial photogrammetric survey is by far the less performing in this case study. This depends mainly on the limited availability of evenly distributed shooting points, and on the large distance between the shooting points and the outcrop, resulting in a significantly less populated point cloud (6.7 M points vs >20 M points). Since the conformation of the site doesn’t allow to distribute better the shooting points, I would suggest adding one or more series of photos shut with longer focal lengths (and using a tripod), to make the resolution of the point cloud comparable with that of the UAV (shot at a much closer distance). Generally, SfM algorithms can combine pictures with different focal lengths, provided that the difference is not too high (about the double – 35 mm with 70 mm etc…).
  • Some authors proposed best practices to mitigate the doming effect of SfM models ( e.g., a second drone flight at the double of the distance, or adding oblique convergent images (James and Robson, 2014)).
  • How do the authors plan to deal with fractures which are visible just as traces, as those shown in Fig. 13C?

Lines 102 – 104: dolomitization is a process occurring before the exposure of carbonate cliffs (during diagenesis or secondary interactions with hot fluids), so I would separate dolomitization from surface dissolution

Lines 133 – 134: please add references for Coltop3D and DSE

Line 137: seem to be > are

Line 150: delete “From a lithologic standpoint”

Lines 151-152: Formation

Fig. 1: please, add the locations of the shooting points of the terrestrial photogrammetric survey.

Fig. 3: the numbers in the reference rectangles are poorly readable.

Fig. 4: add labels a and b)

Fig. 5: the axis labels are not readable, please increase the size of the characters.

Fig. 9: please, indicate the position of the profile in Fig. 8.

Figure 10: show the scale of the contours in the stereoplot.

Figure 10, Tables 6 -9: please, report also the total number of the discontinuities in the  stereoplot and used for the Fisher’s mean and K.

Table 8: is spacing the mean spacing? Is that the spacing calculated over the whole outcrop or only for a smaller area?

Figure 11 is not cited in the text (the topic of the figure is briefly discussed in lines 502 – 507 I guess.

Figure 13: subfigure c) is not commented in the caption.

Line 578: specify which limitations

Line 886: the reference is incomplete

References

James, M. R. and Robson, S., 2014. Mitigating systematic error in topographic models derived from uav and ground-based im-age networks. Earth Surface Processes and Landforms 39(10), pp. 1413–1420.

Author Response

Dear Reviewer,

thank you for your feedback and for your very useful comments and suggestions. 

Here's a summary of our revisions:

1) The topic of terrestrial photogrammetry was also discussed by Reviewer 3. We explained that the aim of this survey was to assess the efficacy of common digital cameras, compared to professional instruments like TLS and drones. This info is in lines 205-209. We added  in Figure 1 the trajectory followed to take the photos with the digital camera. 

2) The paper that you suggested was already in the reference list but I think that there was a little disaster with the plugin that I use for the references. It did not update the reference list after some modifications and, probably, the paper of James and Robson disappeared from the list. I fixed and double-checked all the references.

3) As concerns the fracture traces (Figure 13), we are currently publishing an article dealing with a MATLAB routine developed to perform 2-D quantitative analysis of discontinuties. Being out of the scope of this paper, we did not mention this topic, but, since readers like you could have this question, I added the reference to our paper.

4) We corrected lines 103-104, thank you for noticing that. 

5) We corrected the language errors that you pointed out and added the references for Coltop3D and DSE.

6) Figures 1, 3, 4, 8 and 10 were improved, as well as labels and citations in the text.

7) Unfortunately, the size of the labels of Figure 5 cannot be increased because they were directly provided by CloudCompare. However, the same pictures were put in smaller groups in the supplementary materials, so that the details are visible.

 8) We added the requested data in Tables 6-8. However, we found some difficulties in adding the data in Table 9, as the space is limited. If you agree, we would leave this table as it is, since the useful data are reported in the tables above.

Thank you again for your help!

Sincerely,

Lidia Loiotine

Reviewer 3 Report

The reviewed manuscript presents an interesting comparison between various remote sensing techniques, namely terrestrial laser scanning, terrestrial photogrammetry, and UAV-SfM, and their reliability for geostructural and geomechanical survey purposes. The comparison is performed using both artificial, flat structures (i.e., building walls), and rock mass natural features (i.e., discontinuity planes). The rock mass analysis, in particular, is performed using two software for the automated classification of the point clouds, Coltop3D and the Digital Set Extractor software.

The paper is well written and easy to read, only minor revisions are required, in this reviewer’s opinion, before the manuscript is ready to be published.

The main comment is related to the somewhat scarce “consideration” given to the terrestrial photogrammetry method. There is somewhat a lack of information, provided in the manuscript, about the characteristics of the survey, in terms of camera model, number of stations, distance, and so on. An important point is that 35 mm is quite a short lens, which is generally employed for helicopter or very close range analyses (while the survey distance is unknown in this analysis). Using a longer lens (50-100 mm) would have probably provided higher resolution imagery and, arguably, a denser point cloud. Of course, it is not possible to analyse lots of different camera-lens combinations, but it should be worth mentioning in the discussion that this is an aspect that must be considered, as a function of the site conditions (distance, accessibility, …). I suspect that part of the differences in quality and accuracy observed in this comparison could have been improved by choosing a different lens. Indeed, in some cases, terrestrial photogrammetry can be the only option available for geomechanical survey. For instance at very long range (e.g., > 1km), for which both TLS and UAV may be out-of-range, depending on the model and battery life, respectively.

Having said that, this is a good paper that provides useful insight on the importance of choosing the most appropriate survey method, as a function of the site that needs to be investigated, and its accessibility.

Beside the comment above, which should be addressed by amending the “method section” (to provide additional information), and perhaps the discussion (to address the additional issue of lens selection), I prepared some line-by-line secondary comments that, for the most part, highlight formatting issues or request some more detailed information to be included.

First, in some instances, paper author names are included together with the number in brackets. Should authors use a consistent formatting for in-line references? Eg line 334, 692, 729

Line 111: “especially in places of high tourist attraction”: is this a risk-based consideration (ie, due to the high potential exposure), which can be also valid, for instance, where strategic infrastructure can be present, or merely a third-sector one (ie, to preserve the attractiveness of a place)?

Line 125: mean advantages and drawbacks? Or “main”?

Line 134: please define the acronym DSE at the first use

Line 147: if this is a non-english term, as I suppose, should be in italic, as the one at line 139? I am not sure what author guidelines recommend in such cases.

Line 176: in the inset of Fig 1, the point location of the town is not visible. The scale is also not readable, but this could perhaps be omitted. Also, in the figure there is no mention of the camera stations used for the terrestrial photogrammetry reconstruction. Could this information be included? Also, looking at the morphology of the cliff, and the presence of a single scan position, it seems reasonable to expect quite a bit of occlusions, especially at the sides of the cliff. Would it be possible to include, maybe as an inset or as an additional tab (below the map), a frontal or oblique view of the cliff highlighting (or outlining) the area where in which the comparison was actually conducted?

Line 187: Where were the targets located? Were they on top or at the base of the cliff? Where they “natural points” or actual targets?

Line 195: reference to table 5. Should it be table 4?

Line 198: please be consistent with the capitalization in SFM/SfM. The latter is probably more commonly used.

Line 201: because this paper is about comparison of techniques, it would be useful to include more data in terms of number of stations, location, relative distance, and so on, together with the make of the camera (like it has been done for TLS and UAV)

Line 257: I believe “average distance between points” can be simply called “average point spacing”.

Line 269: It would be good to include in figure 3 photos of the areas where the reference planes were created, to have a sense of the actual roughness of the surface, which I assume would be locally affected by bricks and other asperities (unless the wall is, in fact, smooth)

Line 285: Since the authors mentioned them, what are the main parameters that the algorithm uses?

Line 287: From a grammar perspective, the construction “this allows to [do that]” is incorrect (see here: https://english.stackexchange.com/questions/60271/grammatical-complements-for-allow/60285#60285). The sentence should be rephrased as “Visual inspection […] allowed the identification of zones characterized by higher difference values”, or “Visual inspection […] allowed US to identify the zones characterized by higher difference values”. Same goes for line 628.

Lines 287-289: “More precise detail….grey colorscale”. It is not clear what additional information this procedure may provide.

Line 299: Authors should specify that they are referring to the “point normals”.

Lines 306-308: “The coordinates….representation purposes”. Unclear what authors mean by representation purpose

Line 395: 4 mm should be 4 cm?

Lines 397-398: this seems somewhat a strong statement: terrestrial photogrammetry may be less precise than TLS and UAV-SfM, but is that so to the point that it is inappropriate for geostructural analyses?

Line 399: Graphs in figure 5 are very informative, but it would be easier to compare them if they have the same scale for the x axis (including min and max values), and the 0 was always in the center, also to keep the colors similar (i.e., with a green color at the value 0, and so on). Would it be possible to apply these modifications?

Line 404: the same precision (ie, number of decimal digits) should be used across the table to describe the standard deviation. As it is now, planes 4-6 have double decimal digits than planes 1-3.

Line 468: UAV compared in blue, not red

Line 488: “occlusions” should be mentioned as the reason of missing data

Line 500: I think this observation is very important, and should be stressed: a compass will give us only information on the 20-30 square cm that it covers, while remote sensing will give us data across the full length of the exposed surface, so it is much less affected by undulation and waviness of the discontinuity planes.

Lines 532: It is not easy to understand what color are J1 and J2, and the great circle is a bit too thin to make the color clearly recognizable. A legend would be helpful, and labels for J1 and J2 should be added to the stereonets.

Line 550: Figure 11 is not addressed in the text

Line 598: This figure can be improved. Only tabs a and b are described in the caption, while the only information available to the reader for c and d is that they are taken from the UAV cloud. The caption should be expanded. Also, the figure in the lower right corner seems to be part of d, but in this case there should be some sort of outline that put it together with the figure right above it.

Line 645: this depends very much on the characteristics of the vegetation.

Line 665: “On the other hand”?

Line 722: Here and in some other parts of the manuscript authors refer to “SfM limitations”, but these are never explicitly described.

Line 726: One of the critical aspects to take into consideration is that the ground resolution of digital imagery can be increased by using longer focal lengths. This is almost as important, and perhaps as important, as the choice of camera stations. Especially at longer distances, for which the TLS and UAV may be out-of-range (the latter for battery life issues).

Author Response

Dear Reviewer,

thanks for your positive feedback and suggestions. 

We report the corrections that we made:

1) We specified in the methodology that we used a common digital camera to test it's efficacy compared to professional instruments (TLS, drone). We could not add the exact camera locations because we were constrained by the topography and followed the trajectory (sometimes climbing the subvertical cliff) added in Figure 1. Thank you for explaining the that results of terrestrial photogrammetry could have been improved by using some professional tools, we highlighted this aspet in lines 408-413.

2) We fixed the references in the text and corrected the sentences in lines 11-112, 126, 134-135. For the italic terms, probably they are not allowed from the editors, but we will face this issue during the proof-reading phase.

3) Figure 1: we increased the size of the marker for the town Location in the map and deleted the unnecessary scale. Unfortunately, we were not able to find an appropriate side photo because the morphology is quite complicated to take such a photo. However, we highlighted in red the compared area.

4) we provided the minor revisions (SfM capitalization, specifications on the target used for TLS surveys), lines 257 (now 263), 285 (now 292), 287-289 (now 293-296), 306-308 (now 315-318), 395 (now 405), 397-398 (now 407-413), 404 (table 5), 468 (figure 9), 488 (now 503), 500 (now 516-519), 645 (now 673) , 665 (now 695), 722 (now 751).

5) Figure 3 was improved

6) Unfortunately, we are not able to put the same scales and colours in the small images of Figure 5 because they were directly provided by CloudCompare (for different projects).

7) Figure 10: there's no option to increase the great circles thickness in Dips software. We added the labels "JS1" and "JS2" and put larger figures in the supplementary materials, with the contours legends-

8) Figure 13 was improved.

9) Line 726 (now line 756-757): thank you for the comment, we added this useful info in the text.

We thank you again for your help and support.

Best regards, 

the authors

Reviewer 4 Report

Thank you very much for sending me this text for review. This is a very well preparated technical note with solidly documented methods and the differences between them. The first and most important conclusion is that one TLS scan is not enough...
In the introduction, it is worth writing how this cliff developed in historical times, if there were any large disatsers (rock break. landslides)... if they happened it was probably described in the literature or city chronicles. 
The text clearly shows that the distance (and aspect) to the object is crucial. Can the authors try to specify in the discussion section the maximum distance to scaned object (for the used devices), distance ensuring that the data will be useful?
In tekst are many abbreviations  that should be clarified after first appearing.

Author Response

Dear Reviewer,

thank you for your support. We fixed all the abbreviations and figures/tables labels. Unfortunately, we are not able to provide the maximum distance instrument-target beyond which the results are not reliable. This could be an interesting topic to develop in the future, but probably some preliminary tests at the laboratory scale should be carried out.

Best regards,

the authors